# Application and Progress of Genomics in Deciphering the Genetic Regulation Mechanisms of Plant Secondary Metabolites

**DOI:** 10.3390/plants14091316

**Published:** 2025-04-26

**Authors:** Chong Liu, Hang Xu, Zheng Li, Yukun Wang, Siwei Qiao, Hao Zhang

**Affiliations:** Institute of Special Animal and Plant Sciences of CAAS, Changchun 130112, China; 82101235239@caas.cn (C.L.); 82101222456@caas.cn (H.X.); lizheng@caas.cn (Z.L.); wyk567321@163.com (Y.W.); qiaosiwei1@163.com (S.Q.)

**Keywords:** plant secondary metabolites, genomics, genome, genetic regulation

## Abstract

This review aims to systematically dissect the genetic regulatory mechanisms of plant secondary metabolites in the era of genomics, while comprehensively summarizing the progress and potential impact of genomics in plant secondary metabolism research. By integrating methodologies such as high-throughput sequencing, structural genomics, comparative genomics, and functional genomics, we elucidate the principles underlying plant secondary metabolism and identify functional genes. The application of these technologies has deepened our understanding of secondary metabolic pathways and driven advancements in plant molecular genetics and genomics. The development of genomics has enabled scientists to gain profound insights into the biosynthetic pathways of secondary metabolites in plants such as ginseng (*Panax ginseng*) and grapevine (*Vitis vinifera*), while offering novel possibilities for precise regulation of these pathways. Despite remarkable progress in studying the genetic regulation of plant secondary metabolites, significant challenges persist. Future research must focus on integrating multi-omics data, developing advanced bioinformatics tools, and exploring effective genetic improvement strategies to fully harness the medicinal potential of plants and enhance their capacity to synthesize secondary metabolites.

## 1. Introduction

Plant secondary metabolites (PSMs) are natural organic compounds formed during the long-term evolutionary processes of plants [1]. They play a crucial role in the defense mechanisms and adaptability of plants [2,3]. Additionally, PSMs have wide applications in agriculture, medicine, fragrances, and cosmetics [4]. For example, alkaloids such as morphine and cocaine are indispensable in pain relief, anesthesia, and the treatment of psychiatric disorders [5]. Similarly, compounds like ginsenosides in ginseng exhibit immunomodulatory and anti-stress properties, making them widely used in functional foods and supplements [6,7]. The development and application of these compounds not only promote related industries but also enhance the utilization value of plants [8].

Genomics plays a pivotal role in identifying genes associated with the biosynthesis of PSMs [9]. Through genomics technologies, researchers can precisely locate genes and enzymes closely related to the synthesis of specific PSMs [10]. For instance, high-throughput sequencing and precise chromosome assembly have successfully decoded the ginseng genome, revealing key genes and regulatory elements involved in ginsenoside synthesis [6,11]. After genome assembly, gene function annotation allows for researchers to extract information on secondary metabolite synthase genes, inferring the complete secondary metabolic pathways and providing direction for target gene cloning. Understanding the layout and structure of these genes in the genome has significantly advanced the exploration of gene function and expression regulation mechanisms, laying a solid foundation for identifying PSMs biosynthesis genes and elucidating their evolutionary history [11]. The application of genomics not only improves the accuracy of gene mining but also provides crucial information for subsequent biosynthetic pathway analysis and genetic improvement [9].

Moreover, genomics technologies offer the possibility of gaining a deeper understanding of plant secondary metabolic pathways and their regulatory mechanisms. By validating gene functions one by one and clarifying their roles in metabolic pathways, researchers can implement genetic regulation to optimize metabolic flow, enhance biosynthetic efficiency, and achieve precise regulation of plant secondary metabolism at the genetic level [12]. These research findings not only deepen our understanding of PSM diversity but also significantly accelerate the elucidation of PSM biosynthetic pathways, providing valuable genetic insights for new drug development and the creation of novel agricultural products [13,14].

With the continuous innovation of genomics technologies, the genetic regulatory mechanisms of PSMs have shifted from the periphery to the center of research, becoming a current hotspot. In the historical exploration of PSMs, early research primarily focused on the regulatory effects of environmental factors on metabolic pathways, while the discussion of genetic factors was relatively limited [15]. However, as genetic regulatory pathways gradually became clearer, scientists increasingly recognized the critical role of regulatory factors controlling secondary metabolite synthesis. The rapid development of genomics has provided an opportunity for researchers to delve into plant genomes, accurately identify functional genes closely related to secondary metabolite synthesis, and systematically reveal their complex regulatory networks [2]. The application of genomics technologies has not only greatly enriched our understanding of plant secondary metabolic pathways but also injected new vitality into the fields of plant molecular biology and genomics, significantly advancing the in-depth study of genetic regulatory mechanisms of secondary metabolites. This review aims to summarize how current research progress reveals the genetic regulatory mechanisms of PSMs in the context of genomics and to discuss the key role of genomics in deciphering these mechanisms [12,13].

## 2. Biosynthetic Pathways of PSMs

### 2.1. Major Classes of PSMs

PSMs are small organic molecules that play vital roles in plant adaptation, defense mechanisms, and interactions with the environment [16]. Based on their molecular structures and biological functions, PSMs can be classified into several major categories, including terpenoids, alkaloids, polyphenols, phenylpropanoids, quinones, and enzyme cofactors [17].

Terpenoids are natural products composed of isoprene units, accounting for approximately 40% of plant natural products. They play crucial roles in plant growth, development, and environmental adaptation, such as participating in photosynthesis and respiration, aiding in water and nutrient absorption, and regulating hormone levels in plants [17]. In plant defense mechanisms, monoterpenes and sesquiterpenes, as important components of volatile organic compounds, help plants resist pests and pathogens while also participating in plant-to-plant signaling [18]. For humans, terpenoids are widely used in medicine due to their diverse pharmacological effects, including anti-inflammatory, antioxidant, and anti-tumor properties.

Alkaloids are nitrogen-containing organic compounds widely distributed in the plant kingdom. They play essential roles in plant defense against herbivores and pathogens, protecting plants from damage. These complex cyclic compounds are crucial for maintaining ecological balance in plants. In human medicine, alkaloids such as morphine and cocaine are used for their significant pharmacological effects, including analgesia and anesthesia, though they also carry risks of abuse and dependence.

Polyphenols are a class of secondary metabolites with significant antioxidant activity, including phenolic acids, flavonoids, anthocyanins, and tannins. In plant physiology and defense mechanisms, these compounds protect plants from ultraviolet damage, participate in antioxidant defense systems, and resist pathogens and herbivores, playing a vital role in plant survival and reproduction [19]. For humans, polyphenols such as flavonoids in soybeans not only exhibit antioxidant and anti-inflammatory activities but also contribute to improving nutritional value, preventing cardiovascular diseases, and certain types of cancer [20].

Phenylpropanoids are a class of secondary metabolites derived from phenylpropane units, encompass bioactive compounds such as lignans and coumarins. In plant physiology, lignans have long been recognized as critical secondary metabolites conferring resistance to environmental stresses [21]. For human health applications, lignin derivatives demonstrate therapeutic potential in hepatic disease intervention by suppressing elevated serum levels of alanine transaminase (*ALT*), aspartate transaminase (*AST*), and glutathione S-transferase (*GST*) [22].

### 2.2. Overview of PSMs Biosynthetic Pathways

The biosynthetic pathways of PSMs constitute a complex metabolic network that performs diverse biological functions in plants, including defense, signaling, and environmental interactions [23]. PSMs biosynthesis typically begins with core metabolic intermediates, which are critical precursors in primary metabolism [24]. For example, α-amino acids serve as starting materials for the synthesis of many alkaloids and polyphenols, while acetyl-CoA is a key precursor for terpenoids and flavonoids [24]. Through genomics technologies, scientists can dissect secondary metabolic pathways, identify key enzymes and regulatory factors, and reveal the detailed mechanisms by which primary metabolites are converted into secondary metabolites [25].

#### 2.2.1. Terpenoid Biosynthesis

The biosynthesis of terpenoids relies on two primary pathways to generate isoprene units: the mevalonate (MVA) pathway and the methylerythritol phosphate (MEP) pathway. These isoprenoid precursors polymerize to form monoterpenes, sesquiterpenes, and higher order terpenoids [26]. For example, *SmMYB36* in *Salvia miltiorrhiza* upregulates the expression of key enzymes in both the MEP pathway and downstream tanshinone biosynthesis [27]. Similarly, overexpression of *AaMYB1* in Artemisia annua activates critical genes in the artemisinin biosynthetic pathway, including amorpha-4,11-diene monooxygenase (*CYP71AV1*), amorpha-4,11-diene synthase (*ADS*), farnesyl diphosphate synthase (*FDS*), and aldehyde dehydrogenase (*ALDH1*), thereby enhancing artemisinin production [28].

#### 2.2.2. Alkaloid Biosynthesis

Alkaloids, characterized by at least one nitrogen atom within a heterocyclic ring and alkaline properties, are synthesized via amino acid modifications and intricate molecular rearrangements. A notable example is morphine biosynthesis in *Papaver somniferum*, which requires the sulfotransferase and old yellow enzyme-like reductase fusion—a combination of metabolic phenotype450 (*CYP450*) and oxidoreductase domains—to catalyze the final steps [24]. Liu et al. reported a high-quality chromosome-scale genome assembly of *Coptis chinensis* and identified metabolic phenotype719 (*CYP719*)-encoded (S) canadine synthase as pivotal for berberine biosynthesis [29]. In a breakthrough study, Nett et al. elucidated the complete biosynthetic pathway of huperzine A (*HupB*), involving four enzymes (*SDR-1/2*, *ACT-1*, *CYP782C1*, and *CAL-3*) for 8-carbon substrate formation, two dimerization enzymes (*CAL-1/2*), and four tailoring enzymes (*2OGD-4*, *2OGD-5*, *ABH-1*, and *2OGD-3*) for final structural modifications [30].

#### 2.2.3. Phenolic Biosynthesis

The phenylpropanoid pathway serves as the primary route for synthesizing phenolic compounds. It originates from the shikimate pathway, which converts phosphoenolpyruvate to chorismate. Upon condensation with malonyl-CoA and subsequent modifications (e.g., methylation and glucuronidation), phenylalanine is transformed into representative polyphenols. Polyphenols such as flavonoids and tannins are derived from phenylalanine metabolism through multi-enzymatic reactions and molecular coupling [31]. Investigations into the correlation between polyphenolic metabolites and associated genes in dandelion roots, leaves, and flowers revealed the upregulation of key genes in the polyphenol biosynthesis pathway in floral tissues, and these genes were shown to critically regulate the accumulation of flavonoids and cichoric acid [32]. In *Lamiaceae* plants, functional studies demonstrated that *CYP71D* and short-chain dehydrogenases1 (*SDR1*) participate in the biosynthesis of phenolic monoterpenes, while *CYP76S* and *CYP736A* further catalyze their conversion into thymohydroquinone, thereby elucidating the complete biosynthetic pathway of phenolic monoterpenoids [33].

#### 2.2.4. Phenylpropanoid Biosynthesis

The biosynthesis of phenylpropanoid begins with phenylalanine deamination by phenylalanine ammonia-lyase (*PAL*) to form cinnamic acid, which is subsequently hydroxylated by cinnamate 4-hydroxylase (*C4H*) to yield p-coumaric acid [34]. Flavonoid biosynthesis initiates with the condensation of p-coumaroyl-CoA and three malonyl-CoA molecules, mediated by enzymes including chalcone synthase (*CHS*), chalcone isomerase (*CHI*), and flavonol synthase (*FLS*). Transcription factors such as v-myb avian myeloblastosis viral oncogene homolog (*MYB*), basic helix-loop-helix (*bHLH*), and no apical meristem (*NAC*) coordinately regulate the expression of phenylpropanoid biosynthetic genes. For instance, Han et al. identified 81 enzyme-coding genes critical for coumarin biosynthesis in *Angelica sinensis*, including *C4H*, 4-coumarate (*4CL*) and hydroxycinnamoyl transferase (*HCT*), coumarin synthase, and feruloyl-CoA-6′-hydroxylase (*F6′H*), establishing a foundation for functional validation of these genes [35]. In studies on lignans biosynthesis in *Isatis indigotica*, defective in induced resistance1 (*DIR1*) was identified as a key regulator of the lignans pathway under stress conditions, mediating elevated lignans accumulation to activate stress defense responses [22].

With the rapid development of genomics technologies, we can now gain a deeper understanding of the diversity and complexity of PSM biosynthetic pathways in different plants [36]; the biosynthetic pathway is further divided into different branches under the regulation of different enzymes and genes as shown in Figure 1.

## 3. Genome Assembly and Discovery of Key Genes

### 3.1. Genome Drafts

Before the genomics era, characterizing plant biosynthetic pathways was labor-intensive and time-consuming, often relying on isotope labeling and forward genetics methods, such as inducing random mutants and analyzing their metabolic profiles or cloning biosynthetic pathway genes based on sequence homology to identify single biosynthetic enzymes [37]. The construction of genome drafts relies on sequencing technologies to decode genetic information and predict PSM synthesis pathways on a genome-wide scale. For example, in ginseng, researchers assembled the genome at the scaffold level and annotated homologous genes, identifying 31 genes involved in the mevalonate pathway and 225 members of the UDP glycosyltransferase (*UGT*) family [6]. These studies demonstrate the importance of genome drafts in elucidating PSMs biosynthetic pathways.

With advancements in sequencing technologies and bioinformatics tools, the construction of genome drafts will become faster and more cost-effective [38]. Genome drafts not only enrich our understanding of species’ genetic characteristics but also provide a comprehensive genomic framework for further dissection of PSMs synthesis pathways.

### 3.2. Chromosome-Level Assembly

In genomics research, achieving chromosome-level genome assembly is a major objective, as it provides more precise gene localization and structural variation information, which is essential for a deeper understanding of gene function and regulatory mechanisms. Compared to draft genome assemblies, chromosome-level assemblies offer significant advantages: high-precision assemblies preserve the complete structure of chromosomes [39]; they facilitate the accurate determination of regulatory elements, which is crucial for elucidating the fine-tuned mechanisms of gene expression regulation [40]. These advantages not only enhance the quality and utility of genomic data but also provide richer information for plant secondary metabolic pathways.

Pseudochromosomes are sequences achieved by further assembling scaffolds to the chromosome level, offering a preliminary chromosomal structural framework. Hi-C data assists in the positioning and ordering of scaffolds through three-dimensional spatial information [41], forming structures that approximate complete chromosomes, while Telomere-to-Telomere (T2T) assemblies achieve complete and precise chromosomal assembly through long-read sequencing and high-throughput data integration [42]. The development and application of these technologies are continuously advancing our understanding of plant genome structure and function, providing powerful tools for studying the genetic regulation of PSMs and having profound implications for the genetic breeding of medicinal plants.

In the genome of Melaleuca alternifolia, Zheng et al. discovered through pseudochromosome assembly that the *CYP450* and terpene synthase (*TPS*) gene families are highly correlated with terpenoid accumulation. Genes within the *CYP450* family may influence the composition and diversity of terpenoids by regulating the flow of metabolic intermediates [43]. In the genome of *Acer truncatum*, Ma et al. found that 3-ketoacyl-CoA synthase (*KCS*) genes may contribute to the regulation of nervonic acid biosynthesis, with the *KCS* homologous gene family expanding to 28 members, 10 of which are clustered within a 0.27 Mb region on a pseudochromosome [44,45]. Patel et al. identified 111 *CYP450* genes among the 24,015 annotated genes that exhibited differential expression in roots and leaves, suggesting the presence of 53 *TPS* genes related to terpenoid synthesis in *Andrographis paniculata* [46]. In the study of *Portulaca oleracea*, chromosome-level genome assembly revealed two rounds of whole-genome duplication events, resulting in multiple copies of genes encoding key enzymes/transporters for C4-dicarboxyliccycle (*C4*) photosynthesis and crassulacean acid metabolism (*CAM*), providing insights into the origin, evolution, and metabolic integration of *C4* and *CAM* pathways in *Portulaca oleracea* [47]. Optimizing the utilization of genetic data is of paramount significance in unraveling the biosynthetic pathways by which plants synthesize their bioactive constituents. After this elucidation, the candidate genes implicated in these pathways can be harnessed within the realm of synthetic biology to facilitate heterologous bioproduction. An overview of the key genes associated with a diverse array of recently characterized metabolites and their assembly patterns are presented in Table 1.

## 4. Application of Genomics in the Regulation of Plant Secondary Metabolism

### 4.1. Structural Genomics

Structural genomics plays a pivotal role in the study of PSMs by elucidating the three-dimensional architecture of the genome, thereby uncovering the mechanisms governing gene expression regulation [59]. In the context of PSM biosynthesis, using structural genomics techniques enables researchers to directly observe the precise spatial organization of PSM-related genes within the nucleus [60]. It is critical for understanding the regulatory networks of PSM gene expression, identifying novel biosynthetic pathways, and developing crop varieties with enhanced metabolic traits [61].

#### 4.1.1. Chromosome Conformation Capture (3C)

Chromosome Conformation Capture and its advanced derivatives, such as Hi-C, are powerful tools for investigating the gene regulatory networks of PSMs. These techniques capture the three-dimensional organization of chromosomes, revealing interactions between distant genomic regions that are crucial for understanding long-range gene regulation [62]. In PSM research, 3C technology has been instrumental in identifying genomic regions associated with specific metabolic pathways, including enhancers, silencers, and other regulatory elements that significantly influence gene expression, even when located far from the target genes [63].

The application of 3C has expanded our understanding of PSM gene regulatory networks, particularly in deciphering how complex regulatory elements modulate gene expression. For instance, 3C has demonstrated that under environmental stress, the interaction between promoter regions of specific PSMs genes and distal enhancers is enhanced, leading to the activation or repression of related metabolic pathways [64], providing critical insights into how genes regulate PSMs production in a cell-type-specific manner.

#### 4.1.2. Fluorescence in Situ Hybridization (FISH)

FISH utilizes labeled nucleic acid probes to hybridize with target DNA sequences, allowing for researchers to directly visualize the location of specific genes within cells or tissues [65]. In PSM research, FISH has been employed to localize genes involved in secondary metabolic pathways, providing valuable information on their expression patterns and regulatory networks [66]. Through FISH, researchers have gained deeper insights into the functional and regulatory mechanisms of PSM genes within the plant genome.

FISH has also been instrumental in identifying chromosomal abnormalities and structural variations that impact PSM biosynthesis. For example, FISH can detect copy number variations or chromosomal rearrangements in gene clusters associated with PSM production, which are often linked to plant adaptation to environmental stress or metabolic diversity [67]. FISH mapping reveals dynamic chromosomal rearrangements underlying terpenoid diversity, which promotes the evolution of tomato gene clusters [68]. Jones et al. demonstrate that FISH-based visualization of chromatin architecture uncovers spatial coordination between PSM biosynthetic genes and stress-responsive regulatory elements [69].

### 4.2. Comparative Genomics

Comparative genomics provides a robust framework for dissecting plant secondary metabolic pathways. By comparing the genomes of different species, researchers can identify key genes and enzymes responsible for the synthesis of specific PSMs [70]. For instance, comparative analysis of grapevine genomes has shed light on the evolutionary dynamics of isoflavonoid and flavonoid biosynthesis, while cross-species comparisons have uncovered critical gene families involved in terpenoid biosynthesis [71]. These findings enhance our understanding of the molecular and genetic basis of metabolic diversity across plant species.

#### 4.2.1. Genome-Wide Association Studies (GWAS)

GWAS is a powerful approach for identifying genetic loci associated with specific traits by scanning genetic variations across multiple individuals within a species. It has been widely used to uncover genes influencing PSM biosynthesis [72,73]. For example, GWAS has successfully identified genes responsible for metabolic diversity in rice, providing valuable insights into how plants regulate metabolic pathways through natural variation [74,75]. These discoveries have significant implications for breeding crop varieties with desirable metabolic traits.

In tomato, GWAS has been applied to identify genetic factors controlling fruit metabolic characteristics, advancing research in crop improvement and metabolic engineering [76]. Additionally, GWAS has revealed loci associated with fruit ripening and color formation, contributing to the enhancement of nutritional value and shelf life [75]. As sequencing technologies and statistical methods continue to evolve, GWAS is expected to play an increasingly prominent role in unraveling the genetic regulation of plant secondary metabolism [73].

#### 4.2.2. Bulked Segregant Analysis (BSA)

BSA is a genetic mapping strategy that has proven highly effective in PSM research. By pooling DNA from individuals with distinct phenotypic traits and sequencing the bulked samples, BSA enables rapid identification of genes or genomic regions associated with specific traits [77]. In PSMs studies, BSA has accelerated the discovery of genes controlling the biosynthesis of specific compounds. For example, BSA combined with RNA sequencing has identified the flavonoid 3′,5′-hydroxylase (*F3′5′H*) gene in tea plants, which is linked to catechin concentration [78].

BSA has also been successfully applied in tomato to uncover key genes influencing fruit metabolic traits [77]. A major advantage of BSA is its ability to handle complex genetic backgrounds and reveal genetic variations associated with specific metabolic traits. Furthermore, integrating BSA with transcriptome sequencing (BSA-RNA-seq) has improved the precision of candidate gene identification, as demonstrated in rice for agronomic trait-related genes [79]. This integrated approach has deepened our understanding of the genetic basis of plant metabolic diversity [80]. With the continuous advancement of sequencing technologies and bioinformatics tools, the application of BSA is expected to grow, contributing to precision agriculture and sustainable crop development [81].

#### 4.2.3. Pan-Genomics

Pan-genomics refers to the complete set of genes within a species, encompassing both the core genome (shared by all individuals) and the variable genome (unique to specific individuals). Pan-genomic studies provide critical insights into the diversity of metabolic pathways. For example, pan-genomic analysis of wild and cultivated soybeans has uncovered genetic variations associated with plant metabolism, which may influence environmental adaptation and metabolite synthesis [82].

In *Arabidopsis*, pan-genomic studies have identified compact arrangements of triterpenoid biosynthetic gene clusters associated with chromosomal inversions [82], while in rice, presence/absence variations (PAVs) in key secondary metabolic genes such as terpene synthase28 (*TPS28*), *CYP71Z21*, and *CYP71Z2* have been linked to differences in oryzalexin content among rice varieties [83]. By comparing genomes across individuals, researchers can identify key genes and regulatory elements controlling PSMs biosynthesis, offering valuable genetic resources for crop improvement and drug development. As sequencing technologies advance, pan-genomics is expected to play an increasingly important role in exploring and utilizing plant genetic diversity.

### 4.3. Functional Genomics

Functional genomics aims to elucidate the functions of all genes in the genome and their roles in various biological processes [84]. By integrating transcriptomics, proteomics, and metabolomics data, functional genomics provides a comprehensive platform for understanding the complexity of biological systems [85]. In PSM research, functional genomics has enabled the identification of key genes and regulatory networks controlling the biosynthesis of specific compounds, which have significant applications in medicine, agriculture, and industry [86]. For example, comparing gene expression patterns under different conditions has revealed how plants adjust their metabolic pathways in response to environmental changes, offering valuable insights for improving crop nutritional value and stress tolerance [87].

#### 4.3.1. DNA Microarray

DNA microarray technology is a gene expression analysis tool that allows for researchers to simultaneously monitor the expression levels of thousands of genes. Microarrays enable the comparison of gene expression across different conditions or tissue [88]. In PSM research, microarrays have been used to identify key genes involved in secondary metabolic pathways and their expression changes under environmental stress, shedding light on how plants regulate PSM biosynthesis [89].

For example, DNA microarray analysis has revealed that jasmonic acid treatment significantly upregulates key genes in the artemisinin biosynthetic pathway, such as *ADS* and *CYP71AV1*, leading to increased artemisinin production through genetic engineering [90]. Similarly, in *Catharanthus roseus*, microarrays have identified multiple genes involved in terpenoid indole alkaloid biosynthesis, such as tryptophan decarboxylase (*TDC*), which is regulated by the transcription factor octadecanoid-derivative responsive catharanthus AP3 (*ORCA3*) [91].

#### 4.3.2. RNA Sequencing (RNA-Seq)

RNA-Seq enables analysis of RNA-level changes to study gene expression patterns. It provides critical insights into the regulatory networks of PSM biosynthesis. By leveraging high-throughput sequencing, researchers can identify key genes involved in secondary metabolic pathways and their expression dynamics under different environmental conditions or developmental stages [92].

For example, transcriptome analysis has identified a set of core genes involved in ginsenoside biosynthesis in *Panax ginseng*, offering molecular targets for genetic improvement and enhancing its medicinal value [6]. Similarly, RNA-Seq has been used to identify key genes in tomato flavonoid biosynthesis, such as *CHS*, *CHI*, and *FLS*, enabling the development of high-flavonoid tomato varieties through gene editing [93].

#### 4.3.3. Serial Analysis of Gene Expression (SAGE)

SAGE is a high-throughput technique for quantifying gene expression across the genome. SAGE generates short sequence tags (14–26 base pairs) that represent mRNA molecules, which are then used to screen cDNA libraries for gene identification and quantification [94]. In PSM research, SAGE has been instrumental in identifying active genes during specific biological processes, discovering novel genes, and elucidating their functions and mechanisms [95].

For instance, SAGE has identified key genes in the ginsenoside biosynthetic path-way, such as dammarenediol synthase (*DS*) and cytochrome P450 monooxygenas (*CYP716A4*), which are regulated by transcription factors *WRKY1* and *MYC2* [96]. Similarly, in tea plants, SAGE has uncovered genes involved in catechin biosynthesis, such as *PAL*, *CHS*, and Dihydroflavonol 4-reductase (*DFR*), which are regulated by *MYB* and *bHLH* transcription factor [54]. These findings highlight the utility of SAGE in advancing our understanding of PSM biosynthesis and regulation.

In this section, we provide a concise synthesis of various genomics methodologies that have been employed in the investigation of PSMs. Furthermore, we offer an in-depth analysis of the potential applications, as well as the strengths and limitations, of each technique, with a detailed summary presented in Table 2.

## 5. Functional Validation of Alleles

### 5.1. Yeast Genetic Transformation

Yeast genetic transformation is a fundamental technique in molecular biology and genetics, enabling scientists to introduce exogenous DNA into yeast cells to investigate gene function and metabolic pathways. In the study of PSMs, yeast genetic transformation has been employed for the heterologous expression of plant genes, allowing for functional characterization in non-native systems [97]. This approach has been applied to a wide range of PSMs, including alkaloids, phenylpropanoids, polyphenols, saponins, anthocyanins, and phenolic acids. Such studies validate gene function in yeast, providing critical insights into the biosynthetic pathways of PSMs and facilitating their engineered production. For example, Li et al. achieved high-yield resveratrol production by heterologously expressing resveratrol biosynthetic pathway genes from *Vitis vinifera* in yeast [98]. Through transformed yeast cells, researchers can analyze the impact of nucleotide substitutions on gene expression and protein function, elucidating how such variations influence metabolic pathways and compound biosynthesis [99].

### 5.2. Escherichia Coli Genetic Transformation

Escherichia coli genetic transformation allows for the introduction of exogenous genes or genomic fragments into bacterial cells for functional studies and expression [100]. This technique is particularly valuable for recombinant protein production, gene regulatory network analysis, and metabolic pathway engineering. For instance, researchers have successfully expressed plant-derived isoprenoid biosynthetic pathways in *E. coli*, significantly enhancing the microbial synthesis of pharmaceutically relevant compounds [101]. Additionally, CRISPR-Cas9-mediated genome editing enables precise modifications in *E. coli*, such as targeted adenine-to-cytosine base editing, to investigate the effects of genetic alterations on bacterial metabolic functions [25].

### 5.3. Alternative Techniques for Genotype Modification

A variety of genome editing tools are available for targeted genotype modification to study and improve gene function. Transcription activator-like effector nucleases (TALENs) and zinc-finger nucleases, two early-generation technologies, rely on sequence-specific DNA recognition and cleavage to introduce insertions or deletions at defined genomic loci [102]. These methods have been widely adopted in plant and animal models, including the development of disease-resistant crop varieties and the enhancement of agronomic traits [103]. For example, CRISPR-Cas9-mediated editing of key ginsenoside biosynthetic genes in Panax ginseng has altered the composition of ginsenoside [99].

Base editing technologies, such as base editor systems, enable precise single-nucleotide conversions (e.g., A•T to G•C or vice versa) without inducing double-strand DNA breaks [100]. This approach is particularly suited for studying the functional consequences of point mutations and introducing beneficial allelic variants in crop improvement [104].

Genome simplification and amplification techniques, including multiple annealing and looping-based amplification cycles (MALBAC) and double-stranded annealing motif amplification and sequencing (DAB-seq), offer robust tools for extracting critical genetic information from complex genomes [105]. These methods have been applied in GWAS and population genetics to identify loci associated with crop yield and quality traits [106].

In plants, integrated genomic sequencing and transcriptomic analyses provide deep insights into how genotypic changes influence metabolic pathways and phenotypes [107]. Transcriptomic profiling reveals dynamic gene expression patterns in response to environmental stimuli, while genomic sequencing elucidates the genetic context of these genes [108].

The combined application of these technologies has advanced our understanding of genotype–phenotype relationships, providing powerful tools for crop improvement, therapeutic development, and biotechnological innovation. These advancements not only deepen our knowledge of gene function but also lay the groundwork for precision agriculture and personalized medicine.

## 6. Conclusions

### 6.1. Impact of Genomic Advancements on Plant Secondary Metabolism Research

Advances in genomics have significantly propelled research on plant secondary metabolism, playing a pivotal role in elucidating the biosynthetic pathways and regulatory mechanisms of PSMs. These advancements have not only revealed the genetic basis of plant secondary metabolism but also driven genetic improvements in this field. Genomics encompasses not only gene sequencing technologies but also in-depth studies of genome structure, function, and evolution [109]. Through genomic research, the complexity and diversity of plant secondary metabolic pathways have been unveiled, providing critical insights into the understanding of plant secondary metabolism.

### 6.2. Challenges and Future Directions

Despite significant progress in optimizing PSM production through genetic regulation, numerous challenges remain. For instance, precise control of gene expression to enhance the yield of rare metabolites, or the introduction of novel metabolic pathways to synthesize new compounds, remains unresolved in the field [101]. Additionally, the application of pan-genomics and population genetics will uncover the genetic basis of secondary metabolic diversity, facilitating precision breeding and natural product development [36,72]. Translating this knowledge into plant genetic improvement and medicinal plant breeding may lead to new varieties with enhanced medicinal value and agricultural benefits. Therefore, future research should focus on strategies to improve rare metabolite yields and engineer novel metabolic pathways, emphasizing biosynthetic pathway optimization, innovative genetic modification approaches, multi-omics integration, and genome editing technologies [110]. These efforts aim to accelerate the discovery and utilization of PSMs. Notably, the potential of artificial intelligence and machine learning models in deciphering plant biochemical pathways cannot be overlooked, heralding a new era of precision and efficiency in future research.

## Figures and Tables

**Figure 1 plants-14-01316-f001:**
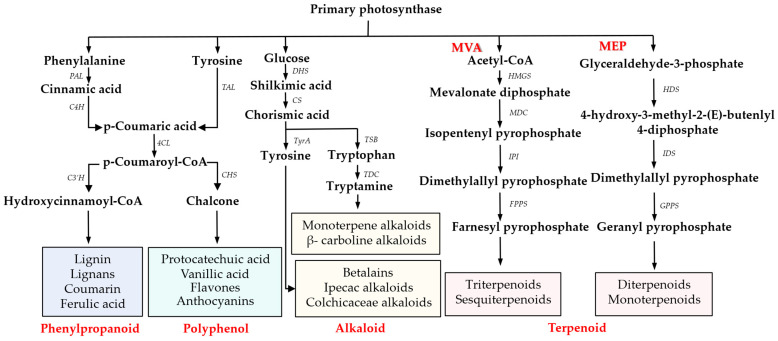
Proposed general biosynthetic pathway of plant phenylpropanoid, polyphenol, alkaloid, and terpenoid. MVA: mevalonic acid; MEP: methylerythritol phosphate.

**Table 1 plants-14-01316-t001:** Sequenced genomes of plants and studying secondary metabolite biosynthesis pathways.

Plants	Assembly Patterns	Gene Assembly	Identified Genes	Biosynthetic Pathways	Reference
*Aralia elata*	PacBio, Illumina, and Hi-C	1.05 Gb	*AeCYP72As*, *AeCSLMs*, and *AeUGT73s*	Oleanane-type triterpenoids	[48]
*Allium sativum*	PacBio, Illumina, and 10x Genomics	16.24 Gb	*AsGSH1b*, *AsGSH2*, *AsPCS1*, *AsFMO1*, and *AsGGT2*	Allicin	[49]
*Acer truncatum*	PacBio, Illumina, 10X Genomics, and Hi-C	633.28 Mb	*KCS11* and *KCS19*	Nervonic acid	[50]
*Coriandrum * *sativum*	PacBio, Hi-C, and 10X Genomics	2.12 Gb	*CsTPS04* and *CsTPS11*	Mannitol, furfural, and linalool	[51]
*Camellia sinensis*	Illumina and Platanus	3.02 Gb	*CsTCS1* and *CsTCS2*	Caffeine and theanine	[52]
*Panax ginseng*	Illumina, PacBio, 10X Genomic, and Hi-C	3.41 Gb	*CYP716A47* and *CYP716A53v2*	Dammarane-type saponins	[53]
*Panax notoginseng*	PacBio, Illumina, and Hi-C	2.66 Gb	*PnUGT1-5*	Ginsenoside	[54]
*Gardenia * *jasminoides*	ONT and Hi-C	534.1 Mb	*CaTCS1, GjUGT75C1*, and *UGT75*	Caffeine and crocin	[10]
*Himalayan yew*	PacBio, Illumina, and Hi-C	10.9 Gb	*CYP725A* and *TRF004A*	Paclitaxel	[55]
*Hypericum* *perforatum*	PacBio, Illumina, and 10X Genomics	393.4 Mb	*HpASMT1* and *HpASMT2*	Melatonin	[56]
*Senna tora*	PacBio, Illumina, and Hi-C	526 Mb	*StPKS1* and *StPKS2*	Anthraquinone	[57]
*Tripterygiumwilfordii*	ONT, Illumina, and Hi-C	340.12 Mb	*TwCYP712K1* and *TwCYP712K2*	Celastrol	[58]

**Table 2 plants-14-01316-t002:** Applications of genomics techniques in studying plant secondary metabolism.

Type	Technique	Application in PSMs Research	Strengths	Limitations	Reference
Structural Genomics	3C	Identifies long-range chromatin interactions regulating PSM gene clusters	Reveals 3D genome architecture; Links spatial organization to metabolic regulation.	Requires high-quality chromatin data; Limited to static snapshots.	[63,64]
	FISH	Localizes PSM biosynthetic genes on chromosomes	Visualizes gene physical positions; Detects structural variations.	Low resolution; Probe design limitations.	[66,67]
Comparative Genomics	GWAS	Discovers loci associated with PSM diversity	Exploits natural genetic variation; High-throughput for complex traits.	Requires large population sizes; Functional validation needed.	[70,71]
	BSA	Rapidly maps genes controlling PSM traits	Cost-effective for bulk samples; Integrates with RNA-seq for precision.	Limited to traits with extreme phenotypes; Misses polygenic effects.	[77,78]
	Pan-genome	Reveals presence-absence variations in PSM genes	Captures species-wide genetic diversity; Identifies novel biosynthetic pathways.	Requires multiple high-quality genomes; Data integration challenges.	[23,83]
Functional Genomics	DNA microarray	Profiles PSM gene expression under stress	High-throughput for known genes; Cost-effective for large-scale studies.	Limited to pre-designed probes; Low dynamic range for rare transcripts.	[10,88]
	RNA-Seq	Identifies dynamic expression of PSM pathways	Detects novel transcripts; Quantitative and strand-specific.	High sequencing depth required; Bioinformatics complexity.	[92,93]
	SAGE	Identifies dynamic expression of PSM pathways	Unbiased for unknown genes; Useful for rare transcript detection.	Labor-intensive library preparation; The efficiency is relatively low.	[94,95]

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
