# Peer review of "Application and Progress of Genomics in Deciphering the Genetic Regulation Mechanisms of Plant Secondary Metabolites"

_plants, 2025, doi:10.3390/plants14091316_

Round 1
Reviewer 1 Report
Comments and Suggestions for Authors
- Line 78-85 is a confusing paragraph. It does not say anything about the title, rather, it is giving instructions of what is expected from the manuscript. The authors should check if this was intended to be in this manuscript.
- Line 158--165 is a very long sentence that can be shortened into two or three sentences. the authors should shorten the sentences.
- Authors should mention the names of the genes in full for the first time, and abbr after. See line 178-183 (CYP, KCS). This should be checked on the whole document.
- Table 1 is not cited in the manuscript, only the contents of it were mentioned.
- Line 209, the authors should not start a sentence with a number, write the in full.
- Line 241-247 is lacking references, authors should include the references for the information and examples given.
- Line 339-356, the names of some genes are abbreviated at the first mention. See comment number 3.
- Line 360-364, where is the evidence of the information provided here? Support with references.
Reviewer 2 Report
Comments and Suggestions for Authors
Dear Authors,
I am sorry to say, but the content of the manuscript is very general, often extensively describing the methodologies of each approach without specific examples of species and secondary metabolites. A more in-depth and detailed study would be needed with its own overview of the potential/effectiveness of each approach, limiting factors, etc.
Chapter 2 is not sufficiently developed. Figure 1 adds no new information and has no substantive value. There are extensive paragraphs in the text without citations (e.g. rows 155-167).
Probably, you forgot to delete the paragraph (rows 78-86) informing about the formal editing of the manuscript. The manuscript contains several formal errors (joined words - tab 1, Latin botanical names without authors, incorrect format for writing Latin botanical species names, reference - row 370, etc.). The formal editing of the manuscript would require more precision.
I am sorry, but the overall impression of the content and form of the manuscript is unconvincing.
Reviewer 3 Report
Comments and Suggestions for Authors
Comments on the manuscript entitled: “Application and Progress of Genomics in Deciphering the Genetic Regulation Mechanisms of Plant Secondary Metabolites”
This review highlights the progress and potential impacts of genomics in understanding plant secondary metabolism. The authors have collected and summarized key informing references. However, some major issues must be addressed to improve the manuscript’s quality.
- Lines 7-8. “This review aims to systematically dissect the genetic regulatory mechanisms of 7 plant secondary metabolites in the era of genomics.”
Unfortunately, no detailed information on the biosynthesis and regulation of PSMs is provided. It is recommended to elaborate on the biosynthesis of each (or major) PSMs, highlighting key biosynthetic and regulatory genes.
- All species names should be put in italics.
- Lines 29-30. “environmental stress … abiotic stresses.” Redundancy.
- Line 56. “genetic leve[12]l.”
- Lines 78-85. Why this comment?
- Section 2.1. It’s important to include lignans in PSMs.
- Section 2.2. A schematic diagram of the biosynthetic pathways of PSMs is required.
- Make sure you cite all Tables and Figures in the manuscript.
- Line 201. “sbiosynthesis,”
- A summary diagram (Figure) for section 4 may improve the readability.
Round 2
Reviewer 2 Report
Comments and Suggestions for Authors
Dear Authors,
thank you for considering my suggestions and for their careful incorporation. Thank you for your cooperation. I wish you much success in your future work.
Author Response
Dear Reviewer,
Thank you sincerely for your thoughtful feedback and for recognizing the efforts we have made to improve the manuscript. We deeply appreciate your expertise and constructive suggestions, which have significantly enhanced the quality of our work.
We are extremely delighted that the revised manuscript has been approved by you.
Once again, thank you for your invaluable guidance. Your insights have not only strengthened this manuscript but also deepened our understanding of plant secondary metabolism.
Wish you all the best in your ongoing work.
Sincerely,
Author: Chong Liu
Reviewer 3 Report
Comments and Suggestions for Authors
I appreciate authors efforts and hard work. They have addressed all issues and improved the manuscript accordingly. However, I would suggest them to revise Figure one. Both polyphenols and phenylpropanoids are synthesize from tyrosine and phenylalanine. So, consider to combine and revise the first two branches.
Author Response
Response to Reviewer
Dear Reviewer,
Thank you sincerely for your thoughtful feedback and for recognizing the efforts we have made to improve the manuscript. We deeply appreciate your expertise and constructive suggestions, which have significantly enhanced the quality of our work.
Response to Comment on Figure 1:
We fully agree with your observation that polyphenols and phenylpropanoids share common precursors (tyrosine and phenylalanine). As suggested, we have revised Figure 1 to integrate these two branches into a unified pathway, clearly illustrating their shared biosynthetic origins and subsequent divergence. The updated figure now:
Combines the polyphenol and phenylpropanoid pathways.
Highlights the role of tyrosine and phenylalanine as precursors, with arrows indicating metabolic flux toward downstream compounds (e.g., flavonoids, lignans, coumarins).
Revised the textual description in Figure 1 to ensure consistency with the sequence of compounds in the image.
Improves visual clarity by reducing redundancy and emphasizing functional connections between enzymes and metabolites.
This revision aligns the figure more accurately with current biochemical understanding and enhances its educational value.
Once again, thank you for your invaluable guidance. Your insights have not only strengthened this manuscript but also deepened our understanding of plant secondary metabolism. We hope the revised version meets your approval and wish you all the best in your ongoing work.
Sincerely,
Author: Chong Liu
Affiliation: Institute of Special Animal and Plant Sciences, Chinese Academy of Agricultural Sciences